# Evaporation of Ti/Cr/Ti Multilayer on Flexible Polyimide and Its Application for Strain Sensor

**DOI:** 10.3390/mi12040456

**Published:** 2021-04-19

**Authors:** Yu-Jen Hsiao, Ru-Li Lin, Hwi-Ming Wang, Cheng-Zhe Cai

**Affiliations:** 1Department of Mechanical Engineering, Southern Taiwan University of Science and Technology, Tainan 710, Taiwan; ruli@stust.edu.tw (R.-L.L.); ma710110@stust.edu.tw (C.-Z.C.); 2Department of Electrical Engineering, Southern Taiwan University of Science and Technology, Tainan 710, Taiwan; tonywang@stust.edu.tw

**Keywords:** Ti/Cr/Ti multilayer, strain gauge, polyimide, evaporation

## Abstract

A flexible Ti/Cr/Ti multilayer strain gauge have been successfully developed based on polyimide substrate. The pure Ti metal strain gauge have shown the hysteresis phenomenon at the relationship between resistance and strain during tensile test. The experimental results of multilayer strain gauge show that adding Cr interlayer can improve the recovery and stability of the sensing electrode. When the interlayer Cr thickness was increased from 0 to 70 nm, the resistance decreased from 27 to 8.8 kΩ. The gauge factor (GF) value also decreased from 4.24 to 2.31 with the increase in the thickness of Cr interlayer from 30 to 70 nm, and the hysteresis phenomenon disappeared gradually. The multilayer Ti/Cr/Ti film has feasible application for strain sensor.

## 1. Introduction

Flexible electronic components (such as strain gauges and tactile sensors) are very important in industrial applications [1,2]. However, the fabrication of complex single-layer and multi-layer structures on polyimide (PI) substrate or the thickness dimensions of film may change its mechanical properties, and related products have been widely used in automatic sensing [3]. Therefore, it is very important to study the mechanical properties of flexible sensing film. In particular, the reliability of products can be longer bending life. Ductile (Pt and Au) films are often used in flexible electronics due to their high conductivity [4,5]. A metallic bilayer film system with a brittle interlayer (Cr and Ti) between a ductile metal film and a polymer substrate is used. Cordill et al. proposed a model that permits the calculation of interface adhesion energy using only the geometry (height and width) of the forming buckles [6,7]. Therefore, we tried to use the combination of Ti and Cr to explore the strain characteristics under tension on the flexible polyimide. Strain gauge is a technique to measure microstrain, which includes complex use of resistance with strain direction. The main working principle of strain gauge is that change in resistance occurs when some materials are under stress [8,9]. Materials usually have different resistance values and can be accurately measured using the Wheatstone bridge circuit [10] at the location where the strain gauge is connected.

Recently, Canavese et al. [11] presented flexible and easy conformable piezo-resistive material composed of Nickel and variable polymer bases with optimized micro-casting and hot embossing techniques and achieved up to nine orders of electrical resistance change when subjected to a mechanical pressure, providing a suitable tactile sensing ability on robot surface. Silver nanomaterials (AgNMs) have also been reported in the flexible strain sensor [12]. It has not only excellent electrical conductivity but also flexibility because of nanoscale size effects. Polycrystalline silicon film was developed on a flexible polyimide substrate using aluminum-induced crystallization process for biomedical pressure and temperature sensing applications [13]. Vaz et al. reported on the development of piezoresistive Ti_x_Cu_y_ thin films, deposited on polymeric substrates (PET) [14]. Based on literature study, works on Ti/Cr/Ti multilayers on polyimide substrate are a few. This fact motivates the current work, which discusses the electrical, mechanical, and force sensing properties of the Ti/Cr/Ti structure. This sandwich structure is similar to the tri-layer films, which are previously studied for mechanical properties [15]. Moreover, the Cr layer has lower resistivity and high adhesion with Ti. Therefore, in this study, Ti/Cr/Ti multilayer films were deposited via E-gun evaporation system to determine the influence of the Cr interlayer on the properties of Ti films. The Ti/Cr/Ti structure has a potential as a sensing electrode in strain gauge applications.

## 2. Materials and Methods

This strain gauge sensor adopts metal as the sensor material, and it uses lift-off patterning process with photomask. Ti/Cr/Ti multilayer films were deposited on a polyimide substrate (HN Kapton) using a Ti metal target (purity: 99.99%) and Cr metal targets (purity: 99.99%) in an evaporation system. The process dimension of the polyimide substrate is 10 × 10 cm^2^. Titanium and chromium metals are placed in two molybdenum crucibles with a capacity of 10 cc, and cross-rotation is used as coating position. The voltage of the electron gun is 6000 V, the current is about 0.04 A, the distance between the evaporation source and the test piece is about 60 cm, and the rotation speed of the stage is 10 rpm. Initially, the evaporating chamber was evacuated to a base pressure of 5 × 10^−6^ Torr with cryo-pump and working pressure of about 5 × 10^−5^ torr. The upper and lower metal (Ti) and intermediate metal (Cr) layer were deposited with the same growth rate of 0.1 nm/s. The thickness of Cr and Ti are Ti/Cr/Ti = 60/30/60, 50/50/50, 40/70/40, and the total metal thickness is about 150 nm. Finally, a layer of SiO_2_ protective layer is attached to the electrode by plasma-assisted chemical vapor deposition (PECVD) system to prevent scratching or external pollution. The process flow is shown in Figure 1.

The structures of Ti/Cr/Ti multilayer films were measured by scanning electron microscopy (SEM, Hitachi SU8000, Tokyo, Japan). Using multi-function power meter (The model is KEITHLEY 2400), the resistance signal value was captured, and then using the computer software (NI LabVIEW), the measurement results were analyzed and displayed. Servo-controlled vertical automatic testing machine (model JSV-H1000), which uses the tension gauge function where its maximum load test can reach 1 kN, along with the software SOP-EG1, can define the test speed, test range, holding time, interval time, and repeatability test.

## 3. Results and Discussion

A miniature strain gauge was created to have a high gauge factor using the Ti/Cr/Ti multilayer films, as shown in Figure 2a. The dimensions of the strain gauge are shown in Figure 2a, the number of loops is 9, the gauge length is 3 mm, and w is the track width, which is 100 μm. The size of a single strain gauge is 13 × 3.5 mm^2^. Figure 2b shows the flexibility test of the strain gauge with the hand. For direct measurement, strain gauge was employed on acrylic beam as shown in Figure 2c. Standard measures were carried out to fix the strain gauge sensor to acrylic beam surface. The standard epoxy was then applied to the acrylic beam surface and sensor and allowed to dry for some time.

The designed strain gauge sensor is an I-shaped structure, and the material used is acrylic. Its thickness is 1 mm, designed middle width is 5 mm, and total length is 45 mm. In this study, ANSYS software was used for simulation analysis and the optimization design of tensile test piece graphics as shown in Figure 3. In the ANSYS software, the material parameter is selected as Poisson’s ratio, which is 0.32, and used a uniaxial force of 45 N for the tensile test. When both sides are fixed and force is applied on one side in the horizontal direction, the strain value at the middle part of the object was measured to be 0.001252. The I-shaped structure of tensile design is used as the base plate for sticking the strain gauge.

The constant cross-section of the strain gauge microsensor of Ti Cr composite metal was photographed by SEM with energy of 5 kV. The structure of the strain gauge microsensor of Ti Cr composite metal is a sandwich structure. The metal used in the first layer and the third layer is titanium with thickness of 47.6 and 47.5 nm, respectively. The metal used in the intermediate layer is chromium with thickness of 46.5 nm. The thickness of Ti/Cr/Ti = 50/50/50 nm can be seen in Figure 4a,b, which shows that there are only two major peaks of titanium and chromium in EDs material analysis of titanium chromium composite strain gauge microsensor.

The relationship between resistance and strain during tensile test of different composite films are shown in Figure 5. All the strain gauge sensors have undergone five different tensile tests with a thickness of 150 nm and the range of strain is from 0 to 0.0034 as shown. In the pure titanium strain gauge, it can be observed that when the tensile specimen is subjected to positive strain in Figure 5a, its resistance value is slowly raised, while on the other hand, when the strain is unloading, its resistance value is slowly returned to the original value, but the rising trend is significantly different from the recovery trend. Thus, it is concluded that the rising trend and recovery trend are nonlinear, and there is an obvious hysteresis loop phenomenon. Figure 5b shows that the total thickness of the strain gauge microsensor of titanium chromium composite metal (60/30/60 nm) is about 150 nm. When the tensile specimen is subjected to positive strain, its resistance value is slowly raised, whereas when strain is unloading, its resistance value is slowly returned to the original value, and the rising trend is similar to the recovery trend, but there are still some microhysteresis loops. Figure 5c shows that the resistance-strain curve of Ti Cr composite metal (50/50/50 nm) is close to a straight line, and the phenomenon of hysteresis loop gradually disappears after the test of strain gauge sensor. When it comes to the Ti–Cr–Ti composite metal (40/70/40 nm), the overlapping resistance-strain straight line can be obtained. It is clear from the Figure 5d that strain gauge sensor showed good resistance-strain properties when the thickness of Cr is increased.

To determine the sensitivity of these strain gauges, the relative change in resistance (ΔR/R) was measured as a function of mechanical strain (ε) in Figure 6. The sensitivity was defined using the so-called gauge factor (GF), which was expressed using the following formula:(1)$$\mathrm{GF}=\left(\frac{\Delta R}{R}\right)/\varepsilon ,$$
where R refers to the initial resistance, ε refers to the mechanical strain of the sensor, and ΔR refers to the amount of change in resistance after the sensor is stretched. After five times forward strain, average of strain gauge sensor is used, the strain factor can be obtained according to the regression curve result calculated from Equation (1) in Figure 6a–d. From the result in Figure 6a, it can be observed that the average regression curve of pure titanium strain gauge is nonlinear, but it is a parabola figure of quadratic program. According to the principle of strain gauge measurement, the differential resistance value must be linear to meet the demand of strain gauge. Therefore, pure titanium is not suitable for strain gauge. Figure 6b shows that the regression curve of strain gauge sensor of Ti/Cr/Ti (60/30/60) composite metal is linear, and its GF value ranges from 4.14 to 4.48. The average value of GF is 4.24. Figure 6c,d shows the average regression curve of different Ti/Cr/Ti composite metals, which also shows a linear change. The average GF of Ti/Cr/Ti (50/50/50) and Ti/Cr/Ti (40/70/40) are 3.38 and 2.31, respectively.

The measurement result of the periodic resistance change of the strain gauge is shown in Figure 7. The resistance value after the maximum strain is about 27.62 kΩ, and the resistance value after recovery is about 27.05 kΩ as shown in Figure 7a. Although the pure titanium resistance can return to its original value, there is no linear rule for the change of resistance with time. In the Figure 7b, Ti/Cr/Ti (60-30-60 nm) means that the thickness of Ti and Cr are 60 and 30 nm, respectively. However, when the strain gauge sensor of Ti/Cr/Ti composite metal is tested for seven times, it can be observed that the maximum resistance value of the tensile specimen under positive strain is almost the same, and the resistance value after unloading strain also returned to the original value. Therefore, the strain gauge sensor of Ti/Cr/Ti composite metal has good recovery, as shown in Figure 7b–d.

The stability of strain gauge sensors of pure Ti (150) and Ti/Cr/Ti (60/30/60) was tested. In Figure 8a,b, when the tensile specimens were subjected to positive strain to the maximum strain and then unloaded strain to no strain, it was stagnant for 100 s. It was observed that the stability of multilayer strain gauge sensor was better than that of pure titanium strain gauge sensor.

It is reported that Ti/Cr interface has good adhesion and can be deposited by vacuum evaporation. According to the literature study, the resistivity of Cr is 1.3 × 10^−7^ Ω m and that of Ti is 4.3 × 10^−7^ Ω m. Cr has a lower resistivity than Ti. In this study, the resistance of pure titanium is about 27 kΩ. When the interlayer Cr thickness was increased from 0 to 70 nm, the resistance decreased from 27 to 8.8 kΩ, as shown in Table 1. Therefore, as the thickness of Cr increases, the resistance trend decreased obviously. On the other hand, the increase in Cr thickness also caused the decrease in GF value. Li et al. studied that the 30 nm Au thin films typically have the resistance of 110 Ω with a gauge factor of 2.6 [16]. In this study, the average GF of Ti/Cr/Ti (50/50/50) is 3.38, which is higher than the gold film.

## 4. Conclusions

In this study, we have successfully developed a flexible multi-layer metal strain gauge based on polyimide substrate. The experimental results of pure titanium single-layer strain gauge show that the change of resistance value is nonlinear compared with the result of initial resistance value and strain, so it cannot be used as a strain gauge. We used the sandwich electrode structure of titanium chromium titanium, fixed its total thickness, and changed the thickness of chromium to find the best parameters. The results show that the recovery test and stability of the electrode are improved after adding Cr. The GF value decreased from 4.24 to 2.31 with the increase in the thickness of Cr interlayer from 30 to 70 nm, and the hysteresis phenomenon also disappeared gradually. It is concluded that it has good strain gauge application with the increase in Cr content.

## Figures and Tables

**Figure 1 micromachines-12-00456-f001:**
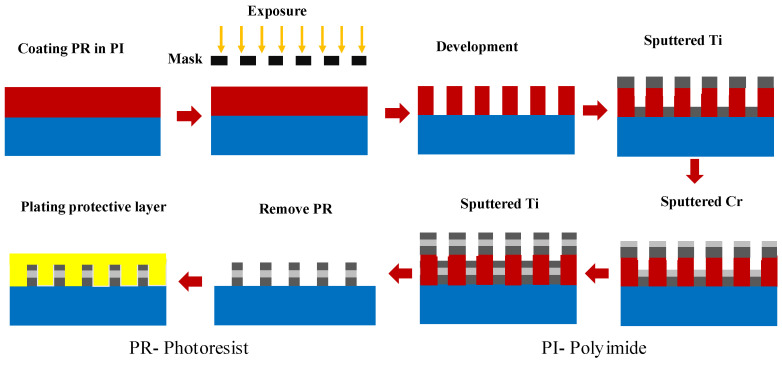
Lift-off process for strain gauge.

**Figure 2 micromachines-12-00456-f002:**
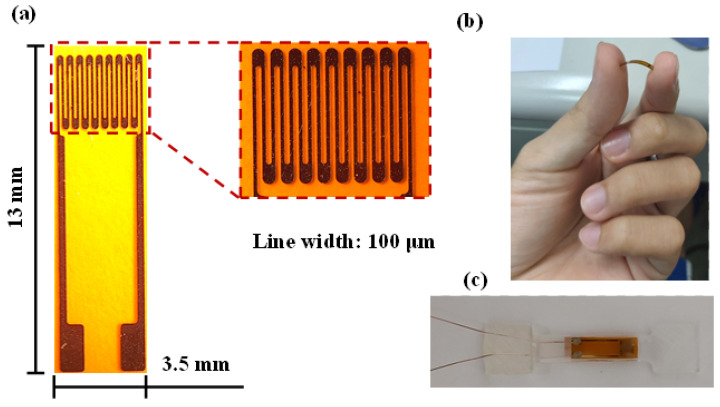
(**a**) Image of a fabricated strain gauge sensor with an area of 13 × 3.5 mm^2^. (**b**) The flexibility test of the strain gauge with the hand. (**c**) The strain gauge sensor employed on acrylic beam.

**Figure 3 micromachines-12-00456-f003:**
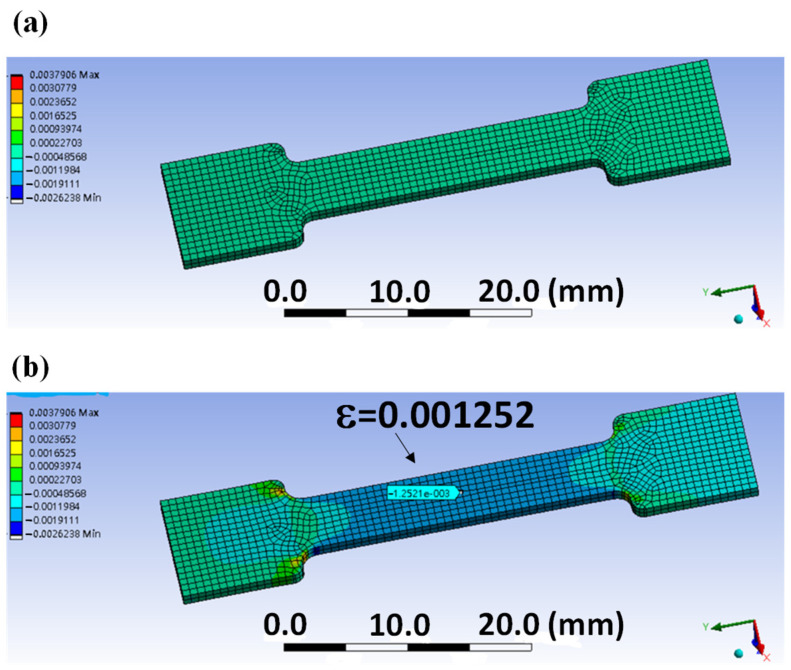
Simulation of the strain gauge (**a**) before and (**b**) after the tension of I-shaped acrylic.

**Figure 4 micromachines-12-00456-f004:**
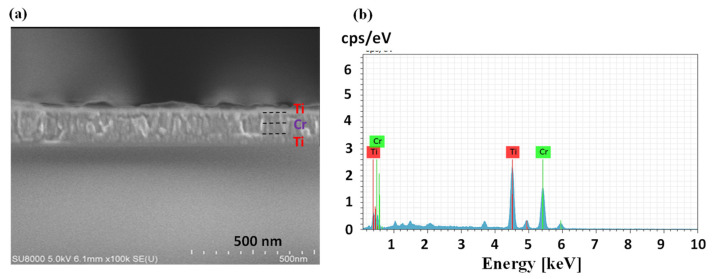
(**a**) SEM image of Ti/Cr/Ti cross-section (50/50/50) and (**b**) EDX analysis of the multilayer film.

**Figure 5 micromachines-12-00456-f005:**
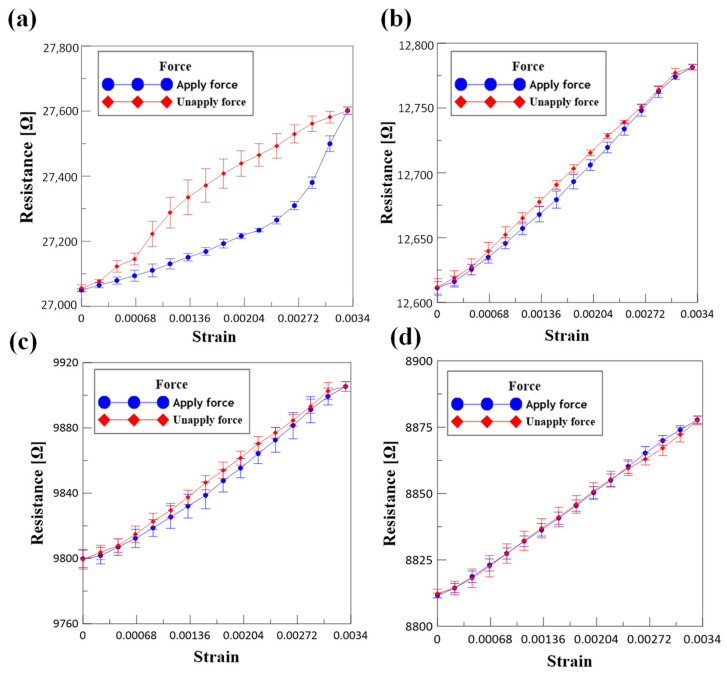
The relationship between resistance and strain during tensile test of different composite films: (**a**) pure Ti, (**b**) Ti/Cr/Ti, (60/30/60), (**c**) Ti/Cr/Ti (50/50/50), and (**d**) Ti/Cr/Ti (40/70/40).

**Figure 6 micromachines-12-00456-f006:**
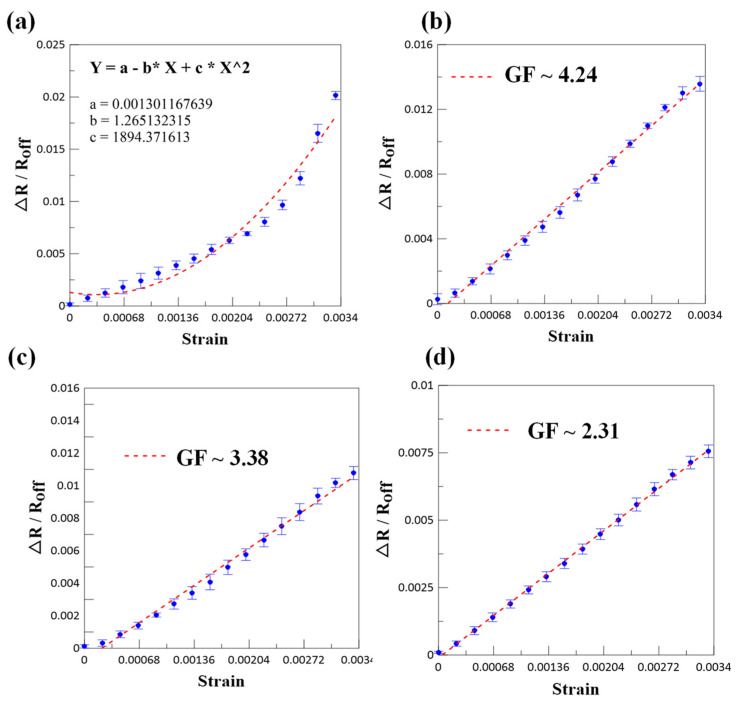
Calculation of the gauge factor (GF) value of strain gauge from the relationship between differential resistance ratio and strain: (**a**) pure Ti, (**b**) Ti/Cr/Ti (60/30/60), (**c**) Ti/Cr/Ti (50/50/50), and (**d**) Ti/Cr/Ti (40/70/40).

**Figure 7 micromachines-12-00456-f007:**
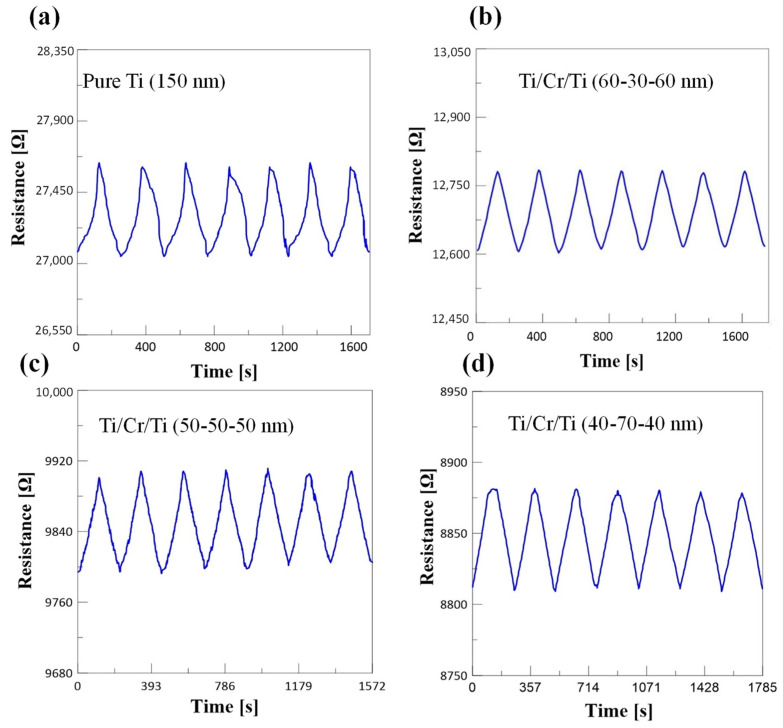
(**a**–**d**) The measurement result of the periodic resistance change of the strain gauge.

**Figure 8 micromachines-12-00456-f008:**
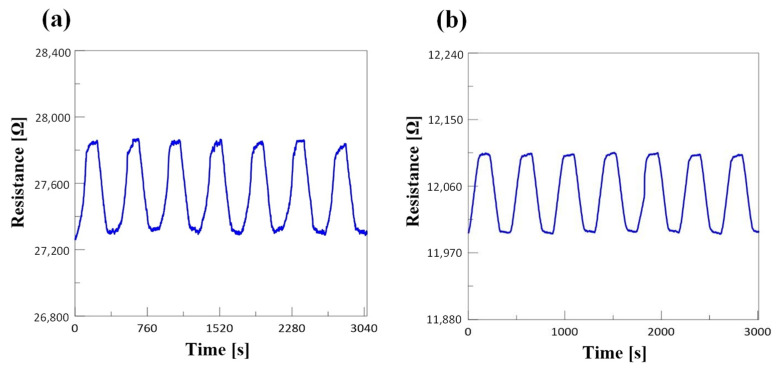
Stability test results of (**a**) pure Ti (150) and (**b**) Ti/Cr/Ti (60/30/60) strain gauge.

**Table 1 micromachines-12-00456-t001:** Resistance and gauge factor (GF) in various sensing material.

Metal Material	Thickness (nm)	Resistance (ohm)	Gauge Factor (GF)
Pure Ti	150	27,050	-
Ti/Cr/Ti	60-30-60	12,610	4.24
Ti/Cr/Ti	50-50-50	9801	3.38
Ti/Cr/Ti	40-70-40	8811	2.31

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
