# Peer review of "Evaporation of Ti/Cr/Ti Multilayer on Flexible Polyimide and Its Application for Strain Sensor"

_micromachines, 2021, doi:10.3390/mi12040456_

Round 1

Reviewer 1 Report

The article by title: "Evaporation of Ti/Cr/Ti multilayer on flexible polyimide and its application for strain sensor" by Authors: Yu-Jen Hsiao, Ru-Li Lin , Hui-Ming Wang , Cheng-Zhe Cai, seems interesting.

It is written concisely. I think it would be worth extending the abstract and introduction.

It would also be worth increasing the number of literature, for example, with a strength analysis of this type of materials.

Please also do a more detailed description and numerical analysis performed in ANSYS software. 

Author Response

Reviewer 1:

  1. It is written concisely. I think it would be worth extending the abstract and introduction.

Answer: We have added some sentences at abstract. In line 14 “When the interlayer Cr was increased from 0 to 70 nm, the resistance decreased from 27 kW to 8.8 kW.” We have also extending sentences at introduction. In line 38 “Canavese et al. [11] presented flexible and easy conformable piezo-resistive material composed of Nickel and variable polymer bases with optimized micro-casting and hot embossing techniques and achieved up to nine orders of electrical resistance change when subjected to a mechanical pressure, providing suitable tactile sensing ability on robot surface.”

[11] Canavese G.; Stassi S.; Stralla M.; Bignardi, C.; Pirri, C.F. Stretchable and conformable metal–polymer piezoresistive hybrid system, Sens. Actuator A Phys. 2012, 186, 191-197. [CrossRef]

  1. It would also be worth increasing the number of literature, for example, with a strength analysis of this type of materials.

Answer: We have added three references [9], [11] and [16].

[9] Kashiri N.; Malzahn J.; Tsagarakis N.G. On the Sensor Design of Torque Controlled Actuators: A Comparison Study of Strain Gauge and Encoder-Based Principles, IEEE Robot. Autom. Lett. 2017, 2, 1186-1194.

[11] Canavese G.; Stassi S.; Stralla M.; Bignardi, C.; Pirri, C.F. Stretchable and conformable metal–polymer piezoresistive hybrid system, Sens. Actuator A Phys. 2012, 186, 191-197.

[16] Li, C.; Hesketh, P.J.; Maclay, G.J. Thin gold film strain gauges, J. Vac. Sci. Technol. A, 1994, 12, 813-819.

  1. Please also do a more detailed description and numerical analysis performed in ANSYS software.

Answer: In the ANSYS software, we selected the material parameter as Poisson's ratio which is 0.32, and used an uniaxial force of 45 N for the tensile test. The information about ANSYS software was added in manuscript from line 97 “In the ANSYS software, we selected the material parameter as Poisson's ratio which is 0.32, and used an uniaxial force of 45 N for the tensile test.”

Reviewer 2 Report

The research presented in this submission is in principle interesting, however the presentation has serious flaws:

  1. The deposition process is not explained in a comprehensive manner.
  2. The SEM micrograph does not show anything because of lack of contrast.
  3. The lettering of figs 4 - 8 is way too small
  4. There are no superscripts and subscripts at places where they should be.
  5. The English is very bad.
  6. There are many typos.

Author Response

Reviewer 2:

The research presented in this submission is in principle interesting, however the presentation has serious flaws:

  1. The deposition process is not explained in a comprehensive manner.

Answer: In this study, the Ti/Cr/Ti multilayer films were deposited on a polyimide substrate (HN Kapton) using a Ti metal target (purity: 99.99%) and Cr metal targets (purity: 99.99%) by E-beam evaporation system. The process dimension of the polyimide substrate is 10 × 10 cm2. To place the titanium and chromium metals in two molybdenum crucibles with a capacity of 10 cc, and coating using position cross rotation. The voltage of the electron gun is 6000 V, the current is about 0.04 A, the distance between the evaporation source and the test piece is about 60 cm, and the rotation speed of the stage is 10 rpm. Initially, the evaporating chamber was evacuated to a base pressure of 5 × 10-6 Torr with cryo-pump and working pressure is about 5´10-5 torr.

The information about the deposition process of Ti/Cr/Ti multilayer films are added in manuscript from line 61Titanium and chromium metals are placed in two molybdenum crucibles with a capacity of 10 cc, cross rotation is used as coating position.  The voltage of the electron gun is 6000 V, the current is about 0.04 A, the distance between the evaporation source and the test piece is about 60 cm, and the rotation speed of the stage is 10 rpm. Initially, the evaporating chamber was evacuated to a base pressure of 5 × 10-6 Torr with cryo-pump and working pressure is about 5´10-5 torr.”

  1. The SEM micrograph does not show anything because of lack of contrast.

Answer: We have re-provided the SEM micrograph with obvious contrast in figure 4(a).

  1. The lettering of figs 4 - 8 is way too small

Answer: Thanks for your kind suggestion. We have enlarged the letters in figure 4-8, as shown below.

Figure 4. (b) EDX analysis of the multilayer film.

Figure 5. The relationship between resistance and strain during tensile test of different composite films (a) pure Ti (b) Ti/Cr/Ti (60/30/60) (c) Ti/Cr/Ti (50/50/50) (d) Ti/Cr/Ti (40/70/40).

Figure 6. Calculation of the GF value of strain gauge from the relationship between differential resistance ratio and strain (a) pure Ti (b) Ti/Cr/Ti (60/30/60) (c) Ti/Cr/Ti (50/50/50) (d) Ti/Cr/Ti (40/70/40).

Figure 7. The measurement result of the periodic resistance change of the strain gauge.

Figure 8. Stability test results of (a) pure Ti (150) and (b) Ti/Cr/Ti (60/30/60) strain gauge.

  1. There are no superscripts and subscripts at places where they should be.

Answer: Thanks for your suggestion. We have checked all superscripts and subscripts in the manuscript.

  1. The English is very bad. There are many typos.

Answer: The manuscript has been re-read carefully and modified with no typo errors. Grammatical and writing style errors in the original version have been corrected by our colleague who is a native English speaker.

Reviewer 3 Report

Dear Authors,

Your paper is interesting, but there are some point sto be improved.

In particular the section "Results and Discussion": The authors report the interesting results obtained, but they do not add any interpretation of the results obtained. In fact the innovation is the use of alternate TiCr/Ti layers, why this composite pile-up works better than pure Ti? They does not try to understand why the use of composite layers improves the stability.

The answer to these questions are not easy and probably will be the focus of another paper. At the moment , probably it will be enough and interesting to the reader to add some table of performance comparison with “standard” gold/platinum or other commercial strain-gauges.

Some suggestions for the paper:

Abstract line 10: Please Prefer impersonal forms and please avoid sentence like this: “We have successfully developed a flexible Ti/Cr/Ti multilayer strain gauge based on polyimide substrate. i.e. write and re-edit: “ A flexible Ti/cr/Ti multilayer strain gauge….. has been developed”.

Another exemplum: “ To our knowledge, works of the Ti/Cr/Ti multilayers on polyimide substrate are few….., please change in, i.e.: “In literature there are few studies focused on…..” and insert some references. “

Line 11:

Attention to the English spelling: i.e. simple typing errors: “The pure Ti metal strain gauge show…” change in: “The pure Ti metal strain gauge shows….”. From this point the English errors will be not outlined, it is demand to the Authors and editors a clear language check.

Figure 1: the caption should describe all the acronyms: recall what is intended with: PR and PI .

Figure 4, left, please use another higher quality/magnification image.

Author Response

Reviewer 3:

Your paper is interesting, but there are some points to be improved.

  1. In particular the section "Results and Discussion": The authors report the interesting results obtained, but they do not add any interpretation of the results obtained. In fact the innovation is the use of alternate TiCr/Ti layers, why this composite pile-up works better than pure Ti? They does not try to understand why the use of composite layers improves the stability. The answer to these questions are not easy and probably will be the focus of another paper. At the moment , probably it will be enough and interesting to the reader to add some table of performance comparison with “standard” gold/platinum or other commercial strain-gauges.

Answer: Thank for your suggest. This topic will be a guide for future work. We have added some sentences at "Results and Discussion" . In line 184 “Li et al. studied that the 30 nm Au thin films were typically 110 Ω with a gauge factor of 2.6 [16]. In this study, the average GF of Ti/Cr/Ti (50/50/50) is 3.38 higher than gold film.”

  1. Abstract line 10: Please Prefer impersonal forms and please avoid sentence like this: “We have successfully developed a flexible Ti/Cr/Ti multilayer strain gauge based on polyimide substrate. ” i.e. write and re-edit: “ A flexible Ti/cr/Ti multilayer strain gauge….. has been developed”.

Answer: Thank you for the suggestion. The sentence has been changes accordingly. “A flexible Ti/Cr/Ti multilayer strain gauge have been successfully developed based on polyimide substrate.”

  1. Another exemplum: “ To our knowledge, works of the Ti/Cr/Ti multilayers on polyimide substrate are few….., please change in, i.e.: “In literature there are few studies focused on…..” and insert some references. “

Answer: Thank you for the suggestion. The sentence has been changes accordingly. “Based on literature study, works of the on Ti/Cr/Ti multilayers on polyimide substrate are few.”

  1. Attention to the English spelling: i.e. simple typing errors: “The pure Ti metal strain gauge show…” change in: “The pure Ti metal strain gauge shows….”. From this point the English errors will be not outlined, it is demand to the Authors and editors a clear language check.

Answer: Thank you for the suggestion. The sentence has been changes accordingly. All grammatical errors have been modified to the best of our knowledge. “The pure Ti metal strain gauge have shown the hysteresis phenomenon at the relationship between resistance and strain during tensile test.”

  1. Figure 1: the caption should describe all the acronyms: recall what is intended with: PR and PI .

Answer: Thank you for the suggestion. PR and PI acronyms have given full names in the figure.

  1. Figure 4, left, please use another higher quality/magnification image.

Answer: We have re-provided higher quality SEM micrograph with obvious contrast in figure 4(a).

Round 2

Reviewer 2 Report

There are still many English mistakes. The native English speaker was not very attentive.

In Figs 3, 4, and 6(a), there is still too small writing.

Not all superscripts have been successfully checked.

Author Response

  1. There are still many English mistakes. The native English speaker was not very attentive.

Answer: Grammatical and writing style errors in the 1st revised manuscript version have been corrected by a native English speaker.

  1. In Figs 3, 4, and 6(a), there is still too small writing.

Answer: Thanks for your kind suggestion. We have enlarged the letters in Figs 3, 4, and 6(a), as shown below.

Figure 3. Simulation of the strain gauge (a) before and (b) after the tension of I-shaped acrylic.

Figure 4. (a) SEM image of Ti/Cr/Ti cross section (50/50/50) (b) EDX analysis of the multilayer film.

Figure 6. Calculation of the GF value of strain gauge from the relationship between differential resistance ratio and strain (a) pure Ti (b) Ti/Cr/Ti (60/30/60) (c) Ti/Cr/Ti (50/50/50) (d) Ti/Cr/Ti (40/70/40).

  1. Not all superscripts have been successfully checked.

Answer: Thanks for your suggestion. We have checked all superscripts in the manuscript. In particular, at line 185 “ According to the literature study, the resistivity of Cr is 1.3 x 10-7 Ω m, and that of Ti is 4.3 x 10-7 Ω m.”

Reviewer 3 Report

The revised version has been improved and now the results are clear.

Author Response

  1. The revised version has been improved and now the results are clear.

Answer: Thanks for the reviewer's suggestion.